# Therapeutic Functions of Stem Cells from Oral Cavity: An Update

**DOI:** 10.3390/ijms21124389

**Published:** 2020-06-19

**Authors:** Ji Won Yang, Ye Young Shin, Yoojin Seo, Hyung-Sik Kim

**Affiliations:** 1Department of Life Science in Dentistry, School of Dentistry, Pusan National University, Yangsan 50612, Korea; midnightnyou@naver.com (J.W.Y.); bubu3935@naver.com (Y.Y.S.); 2Dental and Life Science Institute, Pusan National University, Yangsan 50612, Korea

**Keywords:** dental mesenchymal stem cells, oral cavity, periodontitis, regeneration, immunomodulation

## Abstract

Adult stem cells have been developed as therapeutics for tissue regeneration and immune regulation due to their self-renewing, differentiating, and paracrine functions. Recently, a variety of adult stem cells from the oral cavity have been discovered, and these dental stem cells mostly exhibit the characteristics of mesenchymal stem cells (MSCs). Dental MSCs can be applied for the replacement of dental and oral tissues against various tissue-damaging conditions including dental caries, periodontitis, and oral cancers, as well as for systemic regulation of excessive inflammation in immune disorders, such as autoimmune diseases and hypersensitivity. Therefore, in this review, we summarized and updated the types of dental stem cells and their functions to exert therapeutic efficacy against diseases.

## 1. Introduction

Tissue-specific adult stem cells (ASCs) are the specialized cell population responsible for organ development, homeostasis, and regeneration throughout the lifetime. In general, ASCs have a great self-renew potential with lineage-specific differentiation capacity. For instance, a subset of transplanted hematopoietic stem cells can replenish the whole-blood system of lethally irradiated mice [1]. A single Lgr5+ intestinal stem cell (ISCs) can reconstitute the intestinal epithelial layer containing not only ISC itself but also other mature cell types [2]. Besides organs, connective tissues such as bone and fat are regarded as a rich source of multipotent mesenchymal stromal/stem cells (MSCs). Similar to other organ-derived ASCs, MSCs can self-renew and proliferate well. They are capable of mesenchymal lineage-specific differentiation into bone, adipose tissue, and cartilage both in vitro and in vivo; however, in addition to primary tissue replacement, MSCs are known to play pivotal roles in microenvironment regulation. MSCs contribute to stem cell niche formation and support region-specific ASCs to maintain their stemness and multipotency [3]. They communicate neighbors via direct cell-to-cell contact but also via indirect, secretory factor-dependent signaling so-called paracrine effect. According to the context, MSCs produce a plethora of bioactive molecules that can promote stem/progenitor cell proliferation, determine the direction of differentiation, enhance angiogenesis and even modulate immune responses, leading to wound healing and tissue repair [4,5]. Furthermore, it is widely accepted that MSCs are less immunogenic than other ASCs since they express a low level of MHC antigens and immune cell co-stimulatory molecules [6,7]. In these aspects, MSCs have drawn great interest in the fields of stem cell therapeutics and regenerative medicine.

As connective tissues are widely distributed throughout the body, a variety of MSCs have been described from various origins. Considering that the invasiveness of harvest procedure often limits their clinical utility, easily accessible dental- and periodontal tissues would be an attractive source for the autologous MSC isolation [8,9]. To date, researchers have successfully isolated different types of MSCs from dental specimens that are usually discarded during the treatment such as extracted teeth, attached ligament, and gingival tissue. Of note, dental development begins in the prenatal period but continues beyond the birth until the permanent teeth replace the deciduous ones. For this reason, dental MSCs can be obtained from the developing, immature tissues with higher stemness and flexibility [10]. Dental MSCs share common MSC-related features with regards to self-renewal, extensive proliferation, mesenchymal differentiation capacity, and surface marker expression, while they also exhibit distinctive biological actions depending on their origins [11]. Indeed, they seem to enhance mineral deposition during odontoblast formation and stimulate the neovascularization process within the dental pulp defect. Therefore, the inheritance properties of different dental MSCs should be considered in advance for clinical applications as well as basic science research.

In this review, we aim to provide a comprehensive overview of the general- and unique characteristics of various dental MSCs. This paper also summarizes the latest representative studies showing their regenerative- and immunomodulatory actions in non-dental immunogenic diseases, as well as dental disorders.

## 2. Stem Cells in the Oral Cavity: Sources, General Properties, and Therapeutic Potentials

### 2.1. Stem Cells in Oral Cavity

In terms of developmental view, dental and periodontal tissues are generated via continuous reciprocal actions between ectodermal epithelial cells and ectomesenchymal cells derived from the neural crest as well as the mesoderm [12,13]. While the epithelial progenitors form enamel tissue covering the crown, the ectomesenchyme is responsible for other major compartments of the teeth-dentin, cementum, and dental pulp. Surrounding tissues such as periodontal ligament and gingiva are generated from ectomesenchyme-derived progenitors, implying the presence of ectomesenchymal MSCs. Up to date, various MSC-like cells have been isolated from dental- and periodontal tissues with ectomesenchymal origins including dental pulp stem cells (DPSCs), periodontal ligament stem cells (PDLSCs), gingiva-derived mesenchymal stem cells (GMSCs), dental follicle progenitor cells (DFPCs) and stem cells from apical papilla (SCAP). The naturally exfoliated deciduous teeth also contain dental MSCs, so-called stem cells from human exfoliated deciduous teeth (SHED). Due to their developmental origin, they not only share general aspects of other MSCs but also present neurogenic capacity similar to neural crest-derived stem cells (NCSCs) [14]. In addition, they present distinctive characteristics depending on anatomical locations as summarized below (Table 1).

#### 2.1.1. Dental Pulp Stem Cells (DPSCs)

The dental pulp is a loose connective tissue located in the innermost center part of the teeth. It consists of tiny blood vessels, nerves, and various cells including dental pulp stem cells (DPSCs) [15]. DPSCs are the first discovered MSCs among oral cavity-derived stem cells; in 2000, Gronthos et al. isolated the stem cell-like cell population from the dental pulp of the human third molars, known as “wisdom teeth” [16]. In general, obtained pulp tissue was digested using collagenase type I to generate single-cell suspensions followed by clonal expansion [17].

DPSCs express MSC markers such as CD10, CD13, CD29, CD44, CD73, CD90, CD105, CD146, and CD166 but do not express CD14, CD24, CD34, CD45, and HLA-DR [18]. In addition, they express some neural lineage markers such as Glial fibrillary acidic protein (GFAP), Microtubule-associated protein 2 (MAP2), and Nestin, as well as pluripotent markers Oct-3/4, Nanog and Sox-2 [19,20]. DPSCs can differentiate into odontoblast, osteoblasts, adipocytes, and neural crest cells [21,22]. DPSCs were reported to express the osteoblastic markers such as alkaline phosphatase (ALP), collagen type I (Col I), osteopontin (OPN), and osteocalcin (OCN) and they could differentiate into osteoblast-like cells producing mineralized matrix and give rise to bone tissue [23,24]. Also, DPSCs have superior proliferative- and osteogenic capability compared to bone marrow-derived MSCs (BM-MSCs), suggesting that DPSCs are suitable for tooth regeneration [25,26].

When dental injuries and odontoblast apoptosis occur, rapid activation of DPSCs has been observed followed by their proliferation, migration and differentiation into odontoblast cells, implying that DPSCs play key roles in tooth homeostasis and periodontal regeneration [8,27]. Among the various pathways involved in the odontoblast differentiation, Notch signaling has been reported to induce the odontogenic differentiation of DPSCs and regeneration of the dental pulp [28]. In addition, DPSCs produce neurotrophic factors and promote survival as well as neurite outgrowth of the trigeminal neurons [29]. Of note, DPSCs cultured in the neuronal inductive media condition could express neuronal marker such as nestin, beta III tubulin, neurofilament medium chain (NF-M) and neurofilament heavy chain (NF-H) and differentiate into active neurons [30].

#### 2.1.2. Periodontal Ligament Stem Cells (PDLSCs)

Periodontal ligament (PDL) is one of the major components of the periodontium that connects the tooth-root surface and alveolar bone. The primary role of PDL is tooth support [31]. PDL tissue is composed of heterogeneous cell populations including fibroblast, undifferentiated MSCs, endothelial cells, fibroblast, epithelial cells, the rest of Malassez, and cementoblasts [32,33].

PDLSCs have been identified near the adult PDL tissue of the teeth and play a critical role in periodontal tissue regeneration [34]. The first report describing the isolation and identification of human PDL-derived MSCs was published in 2004 [35]. Since then, there have been continuous attempts to elucidate the function of PDLSCs and their interactions with other dental cells for the clinical application in regenerative periodontics. The PDL cells can be obtained from PDL tissues by enzymatic digestion into single-cell suspensions, which can form adherent clonogenic cell clusters in vitro [36]. Previous studies have reported that PDLSCs represent typical characteristics of BM-MSCs and express MSC-associated cell surface markers such as STRO-1, CD10, CD146, CD13, CD29, CD44, CD59, CD73, CD90, and CD105 while negative for CD14, CD34, CD45, CD38, CD54, or HLA-DR [35,37].

PDLSCs exhibit a self-renewing capacity and possess the multi-potency to differentiate into various types of periodontal tissue components, such as cementum, alveolar bone, and Sharpey’s fibers in vitro and in vivo [38,39]. Several bioactive factors have been reported to regulate the proliferation and differentiation of PDLSCs. A previous study has demonstrated that transforming growth factor-beta (TGF-β1) plays an important role in the fibroblastic differentiation of PDLSCs [40]. TGF-β1 was highly expressed in the entire PDL tissue, and the synthesis of TGF-β1 in PDLSCs depends on their differentiation stage. In addition, the treatment of vascular endothelial growth factor (VEGF) to PDLSCs seem to stimulate the odonto-/osteogenic differentiation in vitro and induce the mineralization of bone structure in vivo. As a powerful inducer of the progenitor cell proliferation in hard tissue regeneration, in contrast, fibroblast growth factors (FGF-2) would enhance the proliferation while inhibiting terminal differentiation of PDLSCs in vitro [41]. Finally, PDLSCs have an immunomodulatory ability comparable to other MSCs; for instance, allogeneic PDLSCs could enhance periodontal regeneration due to their low immunogenicity and immunosuppressive effects on T cell as well as B cells [42].

#### 2.1.3. Gingiva-Derived Stem Cells (GMSCs)

As a part of the periodontium, gingiva tightly surrounds the teeth and underlying alveolar bones to support oral hygiene, providing an intact mucosal barrier against physiobiological challenges with superior wound healing capacity. This soft connective structure consists of the epithelium and lamina propria originated from the outer (perifollicular mesenchyme) and inner layer (dental follicle proper) of the dental follicles [43]. Of note, the gingiva is an easily accessible, highly renewable tissue compared to other dental components and it is often surgically excised during general dental treatment. In this aspect, gingiva has been considered as a convenient source of ASCs in the oral cavity.

Since the first successful isolation by Zhang et al., several protocols have been reported to culture the GMSCs with a description of their characteristics [43,44]. In the immunophenotypic analysis, GMSCs are positive for classical MSC markers including CD29, CD44, CD73, CD90, and CD105 while negative for hematopoietic lineage cell markers such as CD14, CD34, and CD45 [45,46]. They can self-renew while maintaining the long-term proliferative capacity with a faster population doubling time compared to BM-MSCs [47,48].

GMSCs also exhibit a multi-lineage differentiation potential into osteoblast, adipocyte, and chondrocyte as observed in other source-derived MSCs [49]. In addition, evidence of the trans-differentiative potential of GMSCs into β-tubulin^+^ or Glial fibrillary acidic protein (GFAP)^+^ neural lineage cells as well as CD31^+^ endothelial cells have been reported, although it is still controversial and supporting results should be followed to confirm these findings [44,50]. Notably, several attempts have been conducted to regulate their stemness and/or differentiation fate into a designate direction. Ge et al. have evaluated the effect(s) of basic fibroblast growth factor (bFGF) on GMSCs and found that bFGF could not only enhance the proliferative capacity of GMSCs but also stimulate adipogenesis while impeding osteogenesis [51]. It has shown that the encapsulation of GMSCs using hydrogels composed of alginate-gelatin methacryloyl (GelMA) could increase cell viability and control osteogenic potential depending on the hydrogel stiffness [52]. Meanwhile, conditioned medium obtained from embryonic- or postnatal apical tooth germ cell culture seems to support the osteogenic- and odontogenic induction of GMSCs, partially by providing environmental cues found in periodontal development [53,54]. In line with these findings, enamel matrix derivative (EMD) from tooth buds could stimulate cellular proliferation and mineralization during osteogenic differentiation in GMSCs [50].

Similar to other MSCs, GMSCs also play immunomodulatory roles via interacting immune cells [44,55]. Ge et al. have investigated the difference between healthy- and inflamed tissue-derived GMSCs and concluded that those cells might be functionally equivalent in terms of proliferation and differentiation [56]. Given that inflammatory signals could alter the immunomodulatory properties of MSCs, it would be necessary to further evaluate the immune-related phenotypes of GMSCs from inflamed gingiva.

#### 2.1.4. Stem Cells from Human Exfoliated Deciduous Teeth (SHED)

Both SHEDs and DPSCs are derived from the dental pulp; however, SHEDs are derived from deciduous teeth, whereas DPSCs are derived from permanent teeth. Deciduous teeth are the official term for baby teeth, milk teeth, or primary teeth. They start developing during the embryonic stage then begin to come in about 6 months after birth in general.

As first described by Miura et al., the harvesting SHEDs is a simple and convenient procedure with little trauma [57]. Two methods are accepted to isolate and culture of SHEDs: enzymatic digestion and tissue explant. For the enzymatic digestion, the pulp tissues of the deciduous teeth are dissociated with collagenase type I and Dispase for the stem cell isolation. For the latter method, mechanically minced pulps are placed on the tissue culture dishes, allowing the clonal expansion of stem cells [58,59].

SHEDs have a typical fibroblast morphology and express MSC specific markers such as CD45, CD90, CD106, CD146, CD166, and STRO-1 but not the hematopoietic and endothelial markers CD34 and CD31 [60]. As a member of neural crest-derived stem cells, SHEDs also express neural cell markers including Nestin, beta III tubulin, and GFAP some of the pluripotent markers such as Oct4 and Nanog [13,57,61]. SHEDs present multi-lineage differentiation ability to generate osteoblasts, hepatocytes, adipocytes, and neural cells [62,63,64]. SHEDs exhibit increased gene expression patterns of osteocyte markers, including ALP, Col I, and runt-related transcription factor 2 (Runx2) than BMSCs in vitro. In addition, SHEDs can differentiate into functional odontoblasts that secrete mineralizable dentin matrices [65]. Yamaza et al. have demonstrated that the subcutaneous transplantation of SHEDs in nude mice could enhance bone repair via osteoclast inhibition in vivo [66]. SHEDs can also differentiate into vascular endothelial cells that form functional blood vessels by up-regulating the endogenous MEK1/ERK signaling [67].

Due to their origin, SHEDs generally exhibit characteristics similar to DPSCs. However, their proliferation and differentiation capacity are higher than DPSCs and BM-MSCs. Furthermore, SHEDs produce several growth factors, including FGF, TGF-β, connective tissue growth factor (CTGF), and bone morphogenic protein (BMP). In particular, CTGF derived from SHEDs contributes to the neo-dentinogenesis in human corroded teeth [68,69].

#### 2.1.5. Dental Follicle Stem Cells (DFSCs)

Dental follicle (DF) is a loose connective tissue derived from ectomesenchymal tissue and part of the tooth germ, surrounding the enamel organ and the developing tooth [70]. The DF plays various roles during tooth development by regulating the osteoclastogenesis and osteogenesis needed for tooth eruption and tooth-root development of the periodontium [71].

Isolation of human DFSCs has been first established in 2002 [72]. Briefly, DF tissue is dissected from the upside of the dental crown and incubated with a dissociative solution containing collagenase and trypsin. Followed by cell seeding, the subset population of DFSCs would adhere to the plastic dish surface, while non-adherent cells are removed simply by the media change [73]. DFSCs express CD9, CD10, CD13, CD29, CD44, CD53, CD59, CD73, CD106, CD166, and CD271, but not hematopoietic markers including CD 117, CD31, CD34, CD45, and CD133 [14]. Also, DFSCs have putative markers of MSCs and other stem cells such as STRO-1, Oct3/4, Sox-2, Nestin, Nanog, and Notch [70,74].

It has been shown that DFSCs can differentiate into multiple lineage cell types such as osteoblasts, cementoblasts, adipocytes, and neuron-like cells [75,76]. Xiang et al. have investigated the expression pattern of Wnt5a in postnatal DF tissue and its role in the differentiation of DFSCs. They found that Wnt5a could regulate the fate of DFSCs towards differentiation into PDL cells or mineralized cementoblasts and/or alveolar bone osteoblasts [64]. In another study, it is revealed that BMP-2, a member of BMP glycoproteins playing pivotal roles in the development and regeneration of the bone, induces the differentiation of DFSCs into cementoblast/osteoblast-like cells [72]. Similarly, BMP-9 transfected DFSCs present an enhanced osteogenic capacity compared to control cells, suggesting that BMP-9 can be used as osteogenesis inducer of DFSCs [77]. In addition, long-term culture of DFSCs with dexamethasone or insulin could induce the osteoblast lineage commitment and upregulate the mRNA transcripts for osteoblast-specific genes including distal-less homeobox 3 (DLX-3) [78], BMP-2, bone sialoprotein (BSP), and OPN in a culture-duration dependent manner [79]. Yao et al. have demonstrated that dentin matrix acidic phosphoprotein 1 (DMP1) is highly expressed in the DFSCs [80]. The knockdown of DMP1 distinctly reduced the osteogenic capacity of DFSCs, whereas supplementation with mrDMP1 protein to the osteogenic induction medium profoundly increased the osteogenesis, suggesting that DMP1 plays a key role in maintaining and promoting osteogenic capability of DFSCs [80].

#### 2.1.6. Apical Papilla Stem Cells (APSCs)

With the neighboring Hertwig epithelial root sheet, the apical papilla at the apex of developing immature teeth contributes to root development as well as regeneration, implying the presence of resident stem cells. In 2006, it was first reported that the apical papilla is another dental stem/progenitor cell source for so-called ‘stem cells from the apical papilla (SCAPs)’ [81]. SCAPs can be isolated from the apical papilla via single-cell culture after enzymatic dissociation or organotypic tissue culture method [81,82]. The established clonogenic SCAPs have typical characteristics of other dental MSCs in terms of immunophenotype, self-renewal, and differentiation capacity; however, they also exhibit various unique as well as superior properties for clinical application as described below.

First, SCAPs are highly expandable than other dental MSCs [83]. A further study has found that a telomere length of SCAPs is much longer compared to those of DPSCs and PDLSCs because of strong telomerase activity, which enables SCAPs to maintain longer lifespan without premature aging [84]. In addition, SCAPs are relatively resistant to irradiation or chronic periapical inflammation than other dental progenitors, indicating the micro-environmental differences between each stem cell niche [85,86].

Since SCAPs are derived from the immature structure, they express primitive embryonic markers such as Sox2, Oct3/4 and Nanog accompanied with typical MSC markers [81,87]. Among those markers, co-expression of CD146 and STRO-1 is found to be related to early MSCs; indeed, CD146+/STRO-1^+^ SCAPs exhibit superior colony-forming efficiency with increased cumulative doubling than counterpart [88]. CD24, another marker for the pluripotent population, is regarded as a representative surface marker for SCAPs because of its absence in other dental MSCs [89]. It would be worth to indicate that these three markers tend to be declined with passaging, supporting their correlation with superior stemness.

Similar to other MSCs, SCAPs are multipotent and can give rise to mesenchymal lineage cells including adipocyte and chondrocyte; however, SCAPs are known to be rather optimized for osteogenesis and odontogenesis. These findings are not surprising considering that SCAPs are regarded as the precursors for primary odontoblast in vivo [88,90]. Liang et al. found that activation of Wnt/β-catenin pathway could promote osteogenic differentiation [91], while others have reported the opposite results showing the osteogenic/odontogenic potential of Wnt inhibitory molecules such as secreted frizzled-related protein 2 (SFRP2) and Wnt inhibitory factor 1 (WIF1) [92,93]. TGF-β Signaling is another intriguing pathway to regulate SCAP differentiation and increased TGF-β1 activity is correlated to osteogenic suppression of SCAPs [94]. Of note, it is revealed that TGF-β2 is highly up-regulated during osteogenesis and recombinant TGF-β2 could induce a specific increase in odontogenic signatures of SCAPs while it simultaneously reduces osteogenic markers, indicating that each TGF-β family plays some distinctive roles in tooth development and regeneration [95]. Growth factors and hormones can affect osteogenic differentiation of SCAP; the concentrated growth factors from the culture medium, platelet-rich Fibrin, and estradiol have found to contribute to enhanced osteogenesis as well as proliferation [96,97]. Insulin-like growth factor-1 (IGF-1) is another well-described factor that potentiates the osteogenic/odontogenic differentiation in SCAPs [98].

Besides mesenchymal lineage, SCAPs can be differentiated into other lineages; interestingly, sub-population of SCAPs not only expresses neuronal cell-associated markers including βIII tubulin, GAD, nestin, and neurofilament M but also secretes some neurotrophic, neuroprotective factors to support neurogenesis and regeneration [99,100]. The use of fibrin hydrogels or nanomaterial-based scaffolds composed of Graphene and single-wall carbon nanotubes could enhance the neural differentiation capacity of SCAPs [101,102]. In addition, hepatocyte-like cells expressing hepatocyte nuclear factor-1α, alpha-fetoprotein, and albumin could be derived from SCAPs and these differentiated cells exhibit glycogen-storage function and urea production [103]. The angiogenic function of SCAPs also has been proved both in vitro and in vivo, although it is not via direct trans-differentiation of SCAPs into endothelial cells but via promoting migration and tube formation of existing vascular cells [104,105]. Collectively, SCAPs are expected to exert significant roles in tissue regeneration and repair.

### 2.2. Therapeutic Features of Dental MSCs

Tissue regeneration and immunomodulation are the major therapeutic benefits of dental MSCs. Tissue regenerative effect of MSCs can be achieved by (1) the replacement of defected tissues with differentiated dental MSC itself and/or (2) induction of de novo regeneration capacity of endogenous stem cells via remodeling the stem cell niche [106]. Meanwhile, MSCs exhibit an immunomodulatory function by communicating with a variety of immune cells and regulating their activity in a context-dependent manner; mostly, however, MSCs present anti-inflammatory action and stimulate the immune response resolution, which in turn leads to wound healing and tissue repair [107].

#### 2.2.1. Tissue Regeneration

Stem cell therapeutics can be applied to the periodontal treatment for tissue regeneration, which can be achieved by direct cells/tissue replacement or enhancement of endogenous regenerative capacity [108]. During the normal to mild state of the disease, de novo oral cavity-resident stem cells can maintain the tissue homeostasis by itself; however, as the pathologic condition worsens, the exogenous substitutes such as ex vivo expanded/manipulated stem cells would be required to recover the host microenvironment and facilitate the tissue regeneration [109]. In this aspect, dental MSCs can be transplanted into impaired periodontal tissues to function as building blocks to tissue restoration, or stimulate the de novo regeneration via secretion of beneficial factors [110].

Up to date, several studies have proven that dental MSCs can induce the regeneration of not only dental tissues including enamel, dentin, tooth pulp, and PDL but also other organs that share the same developmental origins such as bone and salivary gland [65,111]. Cell modification approaches such as cell immortalization, hypoxic culture condition, and 3-dimensional spheroid formation have been applied to potentiate the stemness and differentiation capacity [111,112,113]. It is further noted that native and synthetic biomaterials can be combined with dental MSCs to provide a physical scaffold that supports engraftment and differentiation of transplanted cells. For instance, when DFSCs contact with aligned PLGA/Gelatin electrospun sheet and fabricated extracellular matrix, periodontium-like complex could be generated both in vitro and in vivo [114]. Another report has shown that transplantation of DFSCs combined with treated dentin matrix into rat alveolar fossa could regenerate the root-like tissue with pulp-dentin complex and PDL connecting cementum-like layer [115]. Huang et al. have successfully generated vascularized dental pulp-like tissue with dentin structure using DPSCs and SCAP mixture seeded in a copolymer of poly-d,l-lactide and glycolide (PLG) scaffold [116]. Collectively, these advanced protocols using modified dental MSCs and biomaterials may provide an optimal approach for regenerative periodontal therapy.

#### 2.2.2. Immunomodulation

Dental MSCs have been reported to present immunomodulatory impact through reciprocal interactions with both innate- and adaptive immune cells as well as the microenvironment, leading to a context-specific immune response regulation [106,117]. For instance, PDLSCs can alter the immune microenvironment to enhance periodontal regeneration by inducing macrophage polarization towards the tissue repair-supportive M2 phenotype while they seem to suppress the proliferation, differentiation, and migration of B cells via enhancing programmed cell death protein-1 signaling, leading to declining in immunoglobulin production [118,119]. DPSCs could impair the viability of natural killer cells and impede their cytotoxic activity in vitro [120]. In addition, SHEDs inhibit the differentiation of T helper 17 (Th17) cells and increase the ratio of regulatory T cells (Tregs) versus Th17 in vivo [66].

Various paracrine effectors mediate the aforementioned immunomodulatory capacity of dental MSCs. When DPSCs are co-cultured with patient-derived peripheral blood mononuclear cells (PBMC), increased secretion of anti-inflammatory cytokines such as prostaglandin E_2_ (PGE_2_), Interleukin-6 (IL-6), and TGF-β could inhibit the proliferation of PBMCs [121,122]. Similarly, GMSCs and PDLSCs can reduce the expansion of PBMCs and allogeneic T cells via the expression of IL-10, IDO, iNOS, and cyclooxygenase-2 (COX-2) [123]. In another study, Fas ligand (FasL) and IL-10 have been suggested as major GMSC-derived factors of T cell inhibition, while others have revealed the importance of PGE_2_, a lipid immune-mediator of both innate and adaptive immune system, and other secretory factors including IL-6 and GM-CSF on immunomodulatory properties of GMSCs [124,125]. Interestingly, preconditioning with pro-inflammatory cytokine interferon-γ (IFN-γ) or hypoxic conditions could promote the immunosuppressive potential of GMSCs both in vitro and in vivo [44,126].

Finally, dental MSCs are known to have low immunogenicity due to the absence of HLA-II DR or T cell costimulatory molecules (CD80 and CD86), indicating themselves as promising candidates for the allogenic stem cell application [127].

## 3. Clinical Application of Dental MSCs

Due to its origins, several attempts have been conducted to apply the tissue regenerative potential of dental MSCs for the dental- and periodontal regeneration (Table 2), while the immunomodulatory capacity of dental MSCs has been evaluated in other immune-related disorders (Table 3) as well as in dentistry.

### 3.1. Dental Stem Cell Application in Dental and Periodontal Disease

#### 3.1.1. Periodontitis

Periodontitis is one of the most common periodontal disease [128]. It begins in the sulcus region of the gingiva then travels to the apex as deposits of plaque and calculus induce inflammation, which slowly disrupts the attachment apparatus [129]. The primary etiology of periodontitis is an ill-defined series of microbial infections [130]. Periodontitis is a slowly developing condition but chronic inflammation induces infectious lesions of the gingiva, PDL alveolar bone, and eventually the tooth loss [131]. Tissue destruction in periodontitis is caused by the breakdown of collagen fibers of the PDL, leading to the periodontal pocket formation between the gingiva and tooth and the resorption of alveolar bone, leading to tooth mobility and subsequent tooth loss [132].

The therapeutic goal of periodontitis is reducing the pocket depth to prevent the progression of the disease and regenerating the lost deprived periodontium [133]. In patients with mild periodontitis, non-surgical therapy such as antimicrobial photodynamic therapy would be sufficient to manage the symptoms, while periodontal surgery is recommended to eliminate the residual periodontitis lesions in severe cases [134]. In general, periodontal surgery would prevent the further progress of the disease [135]; however, it would only stabilize the situations while the tissue loss is irreversible [136].

In this aspect, a multi-level cellular/biomaterial treatment strategy may provide an optimal therapeutic approach for periodontitis. The transplantation of PDLSCs to damaged periodontal tissues would help the remodeling of the local microenvironment as well as tissue by secreting various cytokines that may stimulate the function of residual progenitor cells, which in turn prevents the epithelial down-growth and induces recovery from the periodontal defects [137]. Ding et al. have transplanted cell sheets composed of allogeneic PDLSCs for the treatment of periodontitis in miniature pigs. They found that PDLSC sheets present low immunogenicity and provide appropriate therapy for periodontitis with significant periodontal tissue repair [127]. In addition, stem cells from SHEDs are likely to be a striking candidate for periodontium tissue regeneration. Fu et al. have transplanted SHEDs, composites of allogeneic stem cell sheets derived from minipig deciduous teeth (SPD) combined with hydroxyapatite/tricalcium phosphate (HA/TCP), into a periodontitis model in miniature swine to evaluate their therapeutic potential. The periodontal tissues, including new bone, cementum, and PDL, are regenerated in the periodontal defect area and 75% of the samples had successful furcation regeneration in the SHEDs group, suggesting that SHEDs can provide proper therapy for periodontitis [138]. Furthermore, xenografts of human DPSCs into immuno-deficient athymic mice have shown that DPSCs can produce periodontal cement-like materials, suggesting that DPSCs contain the potential of new therapeutic approaches for periodontal tissue regeneration [139].

#### 3.1.2. Dental Hard Tissue and Pulpal Diseases

Dental caries, also known as tooth decay, is a chronic diseasein the dental hard tissue. Dental caries is a multifactorial disease and caused by localized destruction of dental hard tissues by several risk factors including acid-producing bacteria, fermentable carbohydrates, teeth mineral contents, and oral hygiene [146]. Dental caries arises from an impaired physiological homeostasis between tooth minerals and oral microbial biofilms leading to loss of tooth structure and discomfort [147]. In contrast, dental traumatic injuries are other dental hard tissue defects caused by exogenous damaging factors [136].

Up to now, the main treatment method of dental hard tissue defects is direct filling with amalgam or composite resins to recover the dental defects [148]; however, surgical approaches are often accompanied with remaining problems such as eliminating some healthy tissues to get retention for amalgam or shrinkage, emphasizing the needs of novel therapeutic options [149].

Of note, transplanted DPSCs for the treatment of artificial tooth defects on the premolars and the molars in dogs induced dentin and pulp capping regeneration [140]. Also, DPSCs could reduce dental tissue inflammation and exert immunomodulatory functions either via direct cell-cell contact or via paracrine effects which can prevent proliferation, maturation, cytokine/antibody production and antigen presentation by T cells, B cells, NK cells, and DCs [141].

The pulp tissue reacts in a variety of ways in the presence of periodontal disease [150], while the most common response of the pulp to injury is inflammation so-called pulpitis. In general, root canal therapy has been applied to deal with pulpitis [151]. As the substitution of the naive pulpal tissues with foreign materials leaves the teeth insensitive and changes the composition of enamel and dentin as well as the microenvironment, cell-based therapy is required to regenerate tissues [136]. In this aspect, Iohara et al. have transplanted autologous PDLSCs and progenitors into a root canal with stromal cell-derived factor (SDF-1) after pulpectomy in mature teeth with whole apical closure in dogs. The pulp CD105^+^ cells showed high expression of angiogenic and neurotrophic factors and exhibit complete pulp regeneration [143]. In addition, a previous study has demonstrated complete pulp regeneration by harnessing autologous DPSCs subsets transplanted with SDF-1 in a collagen scaffold into a canine pulpitis model [142].

#### 3.1.3. Alveolar Bone Diseases

Bone loss is one of the hallmarks of periodontal diseases including loss of teeth and chronic periodontitis [152]. Conventional therapies for bone loss include the use of agents such as nonsteroidal anti-inflammatory drugs, glucocorticoids, and anti-rheumatic drugs. These immune regulating therapies may decrease the inflammation temporarily, whereas they do not provide any direct effect on controlling or reversing bone resorption, limiting their uses in the treatment of pathologic bone loss in periodontal defects [153].

Of note, the superior osteogenic regenerative capacity of DPSCs can be applied to bone loss. The treatment of DPSCs with valproic acid can enhance mineralized matrix formation and the expression of bone glycoproteins [154]. DPSCs transplanted with various scaffolds including HA/TCP [155], PLGA [156], collagen [157], nano-fiber hydrogel [158], fibroin [159], and platelet-rich plasma (PRP) [160] to the alveolar bone defects induce bone regeneration. BMP2 and autologous DPSCs promote mineralized tissue formation and osteogenesis [144]. DPSCs in dogs show the highest mean bone-implant contact values than periosteal cells and BM-MSCs [145]. Overall, DPSCs possess a high osteogenic potential and can be applied to the tissue-engineered bone regeneration. 

#### 3.1.4. Application of Dental MSC-Derived Secretome in Dental Disorders

As the paracrine effect is one of the major therapeutic mechanisms of stem cell application, the therapeutic potential of these secretomes derived from dental MSCs on dental disease has been also evaluated [161]. It is revealed that signaling components of the bone morphogenetic proteins (BMPs) and TGF-β pathway seem to be increased in dental MSC-CM. For this reason, CM obtained from dental MSCs is highly-osteogenic and stimulates odontoblast formation as well as dental MSC proliferation compared to CM from other origins [162,163,164]. The ectopic expression of angiogenic and neurogenic factors within the dental MSC-CM would contribute to dental tissue regeneration [165,166,167]. Interestingly, exosomes isolated from DPSC-CM could also exert similar beneficial actions as CM itself in terms of odontogenic induction and dental pulp regeneration [168,169]. Considering their convenience of manipulation, storage, handling, and management compared to whole CM or cells, exosomes could serve as a novel potent option in the dental MSC therapeutic field.

### 3.2. Dental Stem Cell Application in Non-Periodontal Immune-Related Disease

#### 3.2.1. Rheumatoid Arthritis (RA)

Rheumatoid arthritis (RA) is the most common type of autoimmune arthritis affecting about 1% of the world’s population. RA is categorized as an autoimmune disease with unknown etiology that attacks the own tissues due to the disruption of the immune system, leading to joint destruction and deformation. It is noted that inflammatory cytokines contribute to enhanced osteoclastogenesis during RA progression [170]. Several pro-inflammatory cytokines and chemokines are abundant in RA tissues compared to normal, indicating the contribution of abnormal immune responses to the RA progress [171,172,173].

Numerous animal studies have shown that the administration of dental MSCs can improve RA symptoms. It is reported that secretory factors from SHEDs could improve arthritis symptoms and inhibited tissue damage by inducing M2 macrophage polarization and the abrogation of RANKL expression in arthritis model mice [174]. Also, the therapeutic potential of GMSCs have been also demonstrated in RA and other bone erosion related diseases; indeed, systemic infusion of GMSCs can significantly reduce the severity of experimental RA, and restore the balance of Th cell subsets via Fas ligand (FasL) and IL-10 secretion [175]. Transplantation of GMSCs into collagen-Induced arthritis (CIA) mice significantly attenuated the severity of arthritis and osteoclast activity, which in turn reduced bone erosion in vivo via anti-inflammatory CD39-CD73-adenosine signaling pathway [176].

#### 3.2.2. Atopic Dermatitis (AD)

Atopic dermatitis (AD) is a chronic, incurable skin disease that usually begins in infancy or childhood. It is accompanied by itching, dry skin, and eczema. Although the incidence of AD has been increasing worldwide, the pathogenesis of atopic dermatitis has not been completely understood [180].

Various immune cells and related factors are known to be involved in the development of AD. In specific, AD seems to develop when the balance of the Th1/Th2 axis moved towards Th2 response than Th1, following by the B cell induction and inflammatory reactions [181,182,183]. In addition, the infiltration of eosinophils in peripheral blood is elevated in most AD patients. Activated eosinophils increase the production of IL-4, IL-5, IL-6, IL-10, and IL-13 from Th2 cells to increase IgE synthesis [182,184].

Among the dental MSCs, DFSCs show potent immunomodulatory properties which make them attractive approaches for suppression of inflammatory conditions. PBMCs isolated from atopic patients were co-cultured with DFSCs. DFSCs suppressed the expression of IL-4 and proliferation of T cells whereas the expression of IFN-γ and IL-10 was increased. Therefore, DFSCs would exert an immunomodulatory effect and therapeutic outcomes in AD [177].

#### 3.2.3. Multiple Sclerosis (MS)

Multiple sclerosis (MS) is a disease of the central nervous system caused by an autoimmune response to the insulation around myelin in the brain, spinal cord, and optic nerves. In normal conditions, peripheral immune cells do not present in the nervous system due to the blood-brain barrier. In the case of MS, on the contrast, the impaired barrier allows peripheral immune cells to move into the brain and spinal cord of the affected patients [185]. Migrated T cells contribute to the increase of IFN-γ, which induce direct damages to the myelin sheath in the nervous system [186,187]. Eventually, MS can cause permanent damage or deterioration of the nerves.

Recently, it is reported that PDLSCs-derived conditioned medium and purified exosomes reduce pro-inflammatory cytokines such as IL-17, IFN-γ, IL-1β, IL-6, TNF-α, and induce the production of anti-inflammatory IL-10. Therefore, PDLSCs-CM and Extracellular Micro-Vesicles (EMVs) may have a modulatory role against inflammation in MS development [178].

Replacement of damaged neural tissues and cells such as oligodendrocytes are other important options for MS therapy. Of note, DPSCs express specific glial markers, including Olig2, NG2, and O4, and they can differentiate into oligodendrocyte progenitor cells (OPCs) in vitro, suggesting DPSCs as a promising alternative therapy for myelin repair [188].

#### 3.2.4. Sjogren’s Syndrome

Sjogren’s syndrome (SjS, SS) is a chronic autoimmune disease characterized by dry mouth and eyes. The abnormal infiltration of lymphocytes in the salivary gland is often observed in the patient. Following the migration of lymphocytes into the gland, salivary epithelial cells are activated by pro-inflammatory cytokines such as IL-1β, IFN-γ, and TNF-αproduced by immune cells [189].

Both type I and II IFNs have been implicated in the pathogenesis of SS and activation of the IFN-γ receptor causes induction of IFN related genes, promoting antimicrobial protection, apoptosis, inflammation, and tissue damage [190,191]. Type I IFNs can activate B cell-activating factor (BAFF), a powerful regulator of B cell differentiation and proliferation [192]. A recent study showed increased levels of circulating BAFF in patients with SS and up-regulation of BAFF expression in inflamed salivary glands. T, leading to abnormal B cell accumulation within the gland [193,194].

Recently, the possibility of salivary gland regeneration was suggested using DPSCs [179]. In this study, the human salivary gland (HSG) cell line was co-cultured with DPSCs in vitro. It is found that HGSs co-cultured with DPSCs showed the increased expression of lysosomal-associated membrane protein-1 (LAMP-1) and CD44 and increased numbers of acinar structures compared to control. Next, only HSG or combination of HSG-murine DPSCs were subcutaneously transplanted into 2-month-old SS model Rag1 null mice for 2 weeks. Interestingly, co-transplantation of HSG and DPSCs induced significantly higher levels of FGF-7, alpha amylase-1 (AMY-1), von Willebrand factor (vWF), neurofilament-200 (NF-200), VEGFR-3, and VEGF-C than HGS transplantation, indicating that DPSCs can support salivary gland differentiation [179]. 

#### 3.2.5. Inflammatory Bowel Diseases

Inflammatory bowel diseases (IBD) is a chronic disease of the gastrointestinal tract caused by an abnormal mucosal T cell response to commensal and/or pathologic bacterial antigens within the gut lumen. IBD is composed of ulcerative colitis (UC) and Crohn’s disease (CD); CD is regarded as a Th1 mediated inflammatory disorder while UC is regarded as a Th2 mediated disease [195,196]. Indeed, Th2 related cytokines such as IL-4, IL-5, IL-6, and IL-13 are increased in UC patients [197,198]. On the contrary, Th1-related cytokines such as TNF, IFN-γ, and IL-12 are increased in the colonic mucosa of CD [199,200].

Recently, the therapeutic effect of GMSCs in the mouse colitis model [44]. To explore the in vivo immunomodulation capability, GMSCs were transplanted into the subcutaneously dorsal pouches of dextran sulfate sodium (DSS)-treated IBD model mice. As a result, GMSCs significantly ameliorated the colitis symptoms and reduced inflammation via, in part, inducing regulatory T cell response and suppressing the infiltration of pro-inflammatory Th1 and Th17 cells within the colon tissue, implying that GMSCs provide beneficial impact on colitis [44].

## 4. Future Perspectives and Conclusions

So far, stem cells from bone marrow, adipose tissue, and an umbilical cord or cord blood have been intensively studied throughout the world and stem cells from the oral cavity have been relatively less investigated. Therefore, a large body of studies are required for tissue-specific regeneration of dental and oral tissues. Moreover, the unique characteristics and functions of dental MSCs should be precisely demonstrated based on the comparison of dental MSCs with stem cells from other tissues. In addition, future studies should focus not only on the enhancement of transplantation efficiency of differentiated cells by collaborating with biomaterials, 3D printing and drug delivery for regenerative purpose but also on the functional improvement of paracrine functions via novel strategies including priming, genetic modification, surface or conformation engineering of stem cells for immunoregulation.

## Figures and Tables

**Table 1 ijms-21-04389-t001:** Main characteristics of various dental mesenchymal stromal/stem cells (MSCs).

MSC	Origin	First Isolation	Primary Function	In Vitro Differentiation	Main Characteristics
DPSC	Dental pulp of permanent teeth	2000	Dentin formation	Osteoblast Adipocyte Odontoblast Neural crest cells	Express neurogenic markers Express pluripotent markers Higher proliferation capacity and odontogenic capacity than BM-MSCs
PDLSC	Periodontal ligament	2004	Tooth support	Osteoblast Adipocyte Fibroblast Cementoblast	Share similarities with pericyte Expression of PDL-function proteins generate the cementum-PDL structure in vivo
GMSC	Laminar propria of gingiva	2009	Tooth supportWound healing	Osteoblast Adipocyte Chondrocyte Neural cells Endothelial cells	Express pluripotent markers Higher proliferation capacity than BM-MSCs
SHED	Pulpal tissue of deciduous teeth	2003	Secrete mineralizable dentin matrices	Osteoblast Adipocyte Odontoblast Neural cells Hepatocyte Endothelial cells	Express neurogenic markers Express pluripotent markers Higher proliferation capacity than DPSCs and BM-MSCs
DFSC	Dental follicle	2002	Tooth development	Osteoblasts Adipocyte Cementoblast Neural cell	Higher immunomodulatory effects than other ASCs Regulate the osteoclastogenesis and osteogenesis
APSC	Apical papilla of an immature tooth root	2006	The primary source of odontoblasts at root region	Osteoblast Adipocyte Odontoblast Neural cells Hepatocyte	Express pluripotent markers Highly expandable than other dental MSCs

**Table 2 ijms-21-04389-t002:** Therapeutic application of dental MSCs to induce dental regeneration.

MSC Type	Model/Condition	Injection Route	Main Effects	Ref.
DPSC	Immuno-deficient athymic mice	Xenograft	Differentiated into cementoblast-like cells Differentiated into adipocytes and collagen forming cells	[139]
	Tooth defects in beagle dogs	Autograft	Dentin regeneration ↑ Pulp capping regeneration ↑	[140]
	Begie XIDIII nu/nu mice	SC injection	Dental tissue inflammation ↓ Pro-inflammatory cytokine secretion ↓	[141]
	Canine pulpitis model	SC injection	Dental pulp regeneration ↑ Present angiogenic/vasculogenic/neurogenic potential	[142]
	Apical closure in dogs	Autologous transplantation	Highly expressed angiogenic/neurotrophic factors Neovascularization ↑ Dental pulp regeneration ↑	[143]
	Aveolar bone defects in New Zealand rabbit	Autologous transplantation	Alveolar bone regeneration ↑ Expressing STRO-1 and vimentin	[144]
	Bone defects in dogs	Autologous transplantation	Osteogenic potential ↑	[145]
PDLSC	Periodontitis in Miniature Swine	Flap surgery	Periodontal tissue regeneration ↑ Alveolar bone regeneration ↑ T Cell Proliferation ↓	[137] [127]
SHED	Periodontitis in Miniature Swine	Flap surgery	Periodontal bone regeneration ↑	[138]

**Table 3 ijms-21-04389-t003:** Immunomodulatory capacity of dental MSCS targeting autoimmune diseases.

MSC Type	Target Disease	Model	Injection Route	In Vitro Effects	In Vivo Effects	Ref.
SHED (CM)	RA	CIA mouse	I.V	M2 polarization↑ RANKL-induced Osteoclastogenesis↓	M2 polarization↑ M2 cell markers↑ RANKL expression↓ Osteoclastogenesis↓ Swelling↓	[174]
GMSC	RA	CIA mouse	I.V	Induces T cell apoptosis AnnexinV^+^, 7AAD^+^↑ Th1/Th17 polarization↓ Treg polarization↑	IgG, TNF-α expression↓ Immunosuppressive↑	[175]
GMSC	RA	CIA mouse	I.V	NF-κB P65/P50↓ Osteoclastogenesis↓	Th1/Th17 polarization↓ TNF-α expression↓ Pro-inflammatory cytokine↓ RANKL expression↓ Osteoclastogenesis↓ CD39, CD73, TGFβR1, MAPK expression↑	[176]
DFSC	AD	patients PBMCs	–	CD4^+^, CD8^+^ T cell proliferation↓ IL-4 expression↓ IL-10 expression↑	–	[177]

PDLSC (CM)	MS	MS	I.V	–	Body weight↑ IL-17, IFN-γ, IL-1β, IL-6, TNF-α expression↓ IL-10/TGF-β expression↑ STAT1, p53, caspase-3, Bax expression↓	[178]
DPSC	SS	Rag1 null mice	S.C.	Acinar differentiation↑	FGF-7/10, NF-200, VEGFR-3, VEGF-C, AMY-1 level↑	[179]
GMSC	IBD	DSS model	S.C.	PBMC proliferation↓ IDO, IL-10 expression↑	IL-10, FoxP3 expression↑ CD4^+^ T cell infiltration↓ Treg cell infiltration↑	[44]

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
