# Peer review of "Therapeutic Functions of Stem Cells from Oral Cavity: An Update"

_ijms, 2020, doi:10.3390/ijms21124389_

Round 1

Reviewer 1 Report

Dear authors

the current review entitled "Therapeutic functions of stem cells from oral cavity: an update" by Yang JW et al. is very interesting and gives a significant contribution to the scientific community in the field of stem cell functions as therapy in the oral pathologies. I found the paragraph 3.1.1 too long compared to the applications in the periodontitis pathology. I suggest to review this section. The real weak point of the paper is represented by bibliography. It must be revised. There are references without the journal's name, without number page, in some reference the authors are missing (217), etc, etc. Moreover, in my opinion, is too long. I advise you to review it, also taking into consideration that at least 50% of the bibliographies should be from the last 5 years. 

please see the instructions for authors

Author Response

We thank this reviewer for the positive evaluation. Based on the suggestion, we thoroughly reviewed the manuscript and revised some sections including 3.1.1 concisely. Grammatical errors as well as inappropriate expressions were also corrected. Also, we found that there have been unintended errors while editing the reference section. In this revised work, we carefully checked the reference styles and updated citations to reflect the current status of this field. And the number of total references has been reduced (more that 50 removed in overall). Thus, we believe that the revised manuscript is now improved and well-organized.

Sincerely,

Hyung-Sik Kim

Reviewer 2 Report

Congratulations. The article is a very thorough and orderly review.Tables add substantive value, where the authors analyze researches on the use of stem cells for various indications.However, there is a lack of descriptions for the tables, which would help the reader to understand them. Besides, I have no comments.

Author Response

We thank this reviewer for the positive evaluation. As suggested by the reviewer, we added table titles and re-arranged the format of tables to improve the readers` understanding. We also reviewed the whole manuscript carefully and corrected grammatical errors and typos. We hope this revised work will meet reviewer's expectation.

Sincerely,

Hyung-Sik Kim